# Holographic codes from hyperinvariant tensor networks

Matthew Steinberg[1,2], Sebastian Feld[1,2] & Alexander Jahn [3,4] ✉

Holographic quantum-error correcting codes are models of bulk/boundary dualities such as the anti-de Sitter/conformal field theory (AdS/CFT) correspondence, where a higher-dimensional bulk geometry is associated with the code's logical degrees of freedom. Previous discrete holographic codes based on tensor networks have reproduced the general code properties expected from continuum AdS/CFT, such as complementary recovery. However, the boundary states of such tensor networks typically do not exhibit the expected correlation functions of CFT boundary states. In this work, we show that a new class of exact holographic codes, extending the previously proposed hyperinvariant tensor networks into quantum codes, produce the correct boundary correlation functions. This approach yields a dictionary between logical states in the bulk and the critical renormalization group flow of boundary states. Furthermore, these codes exhibit a state-dependent breakdown of complementary recovery as expected from AdS/CFT under small quantum gravity corrections.

The field of quantum error correction, while relevant for many practical applications in the context of quantum computation, also has deep connections to high-energy theory and quantum gravity. This is exemplified by the anti-de Sitter/conformal field theory (AdS/CFT) correspondence, a conjectured duality relating $d+1$-dimensional bulk quantum gravity on an asymptotically AdS space-time background to a $d$-dimensional CFT on its boundary[1,2]. This duality implies a dictionary between operators and fields between these two theories, and the details of this dictionary exhibit the defining features of a quantum error-correcting code[3]. Concretely, AdS/CFT relies on a parameter $N$ that characterizes both theories: While it counts the number of degrees of freedom of the boundary CFT, in the bulk the value of $N$ determines the effective gravitational strength in terms of the gravitational constant $G \sim 1/N^2$ (in units where $\hbar = c = 1$) and thus the quantum-ness of the bulk: In the $N \to \infty$ limit, the bulk theory is merely semi-classical gravity, while finite values of $N$ imply quantum gravity corrections. Although non-perturbative quantum gravity is poorly understood, the most popular setting of AdS/CFT is in this $N \to \infty$ limit, potentially including perturbative corrections. It is this limit in which the code structure of AdS/CFT becomes most apparent: Counter-

intuitively, the number of degrees of freedom of a bulk field on a semi-classical AdS background, when restricted onto a chosen time-slice at time $t$ and made finite via discretization and a radial cutoff, is smaller than that of the boundary CFT state. This is because considering only bulk states on a fixed semi-classical geometry imposes a restriction on the boundary Hilbert space, leaving out CFT states that are dual to a non-geometrical bulk or contain strong back-reaction (such as black hole states). As a result, the AdS/CFT dictionary becomes an (approximately) isometric code between bulk and boundary: The logical space of states associated with a semi-classical AdS geometry are encoded in a code subspace of the boundary CFT states[3]. This encoding has peculiar geometric features: As shown in Fig. 1a, a region $A$ of the boundary CFT can be associated with an *entanglement wedge* $a$, bulk information in which can be fully represented on $A$. Conversely, any local bulk information around a point $x$ can be represented on any boundary region whose entanglement wedge contains it[4,5]. For a bipartition $\mathcal{H} = \mathcal{H}_A \otimes \mathcal{H}_{A^c}$ of the boundary Hilbert space (which is finite-dimensional due to the bulk discretization and cutoff), any such local bulk information can be represented on $A$ or $A^c$, but not both, a feature known as *complementary recovery*. In the bulk, $a$

[1]QuTech, Delft University of Technology, 2628 CJ Delft, The Netherlands. [2]Quantum and Computer Engineering Department, Delft University of Technology, 2628 CD Delft, The Netherlands. [3]Department of Physics, Freie Universität Berlin, 14195 Berlin, Germany. [4]Institute for Quantum Information and Matter, California Institute of Technology, Pasadena, CA 91125, USA. ✉e-mail: a.jahn@fu-berlin.de

**a**

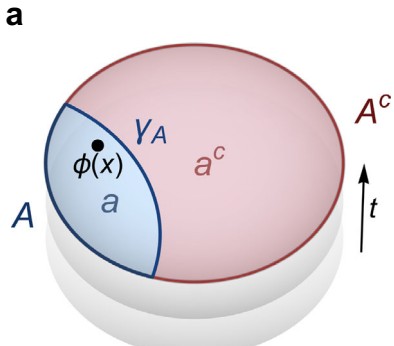

**b**

**Fig. 1 | Complementary recovery in holography. a** In continuum AdS/CFT on slices at constant time $t$, a bipartition of the boundary CFT into two regions $A$ and $A^c$ is equivalent to a bipartition of the bulk into two entanglement wedges $a$ and $a^c$, separated by a Ryu–Takayanagi surface $\gamma_A$. Any bulk (field) operator $\phi(x)$ can be fully reconstructed on either $A$ or $A^c$, but not both. **b** In the holographic tensor network code introduced here, a boundary bipartition leads to bulk wedges that are separated by a large residual region (white); an operator $\phi$ in this region cannot generally be reconstructed on either $A$ and $A^c$.

and $a^c$ are separated by the Ryu–Takayanagi (RT) surface $\gamma_A$, a geodesic (extremal surface in higher dimensions) whose area determines the dominant $O(N^2)$ part of boundary entanglement entropy $S_A \equiv S[\rho_A] = -\operatorname{tr}_A(\rho_A \log \rho_A)$ at large $N$[6], where $\rho_A = \operatorname{tr}_{A^c}(\rho)$ is the reduced density matrix of subregion $A$. Explicitly including $O(N^0)$ corrections, it is given by[7–9]

$$S_A = \frac{\operatorname{area}(\gamma_A)}{4G} + S_a + O(G), \qquad (1)$$

where $S_a$ is the bulk entropy between $a$ and $a^c$. As we consider quantum effects in the bulk, area($\gamma_A$) should formally become an expectation value of an area operator. It has been conjectured that (1) becomes exact in all orders of $G$ if one replaces $\gamma_A$ by a *quantum extremal surface*, extremizing the entire (quantum) entropy rather than the classical area[10]. The general form of (1) is a direct consequence of the holographic code properties[11].

Both complementary recovery and a form of (1) can be readily reproduced in discrete toy models of holography, most notable the family of perfect holographic codes, also known as HaPPY codes after their creators[12]. These codes are based on tensor networks of so-called perfect tensors arranged on a regular hyperbolic lattice, producing a code between logical qubits within the bulk of the tensor network and physical qubits on its boundary. In this discretization, the Ryu–Takayanagi surface $\gamma_A$ becomes a cut through the tensor network. The bulk area term $S_a$ in (1) becomes nonzero once the logical state in the bulk contains entanglement between $a$ and $a^c$. HaPPY codes can be constructed on various hyperbolic tilings and for higher local dimensions (i.e., with qudits instead of qubits). While this model reproduces the quantum error correction properties of AdS/CFT up to bulk discretization artifacts, the boundary code space does not contain states that can be readily associated with physical CFT states. While the entanglement entropy scaling agrees with results for critical states[12,13], the expected smooth polynomial decay of $n$-point correlation functions with distance is precluded by the code properties of the model; for example, simple spin-spin correlation functions such as $\langle X_j X_k \rangle$ of Pauli $X$ operators between sites $j, k$ always vanish in the {4, 5} pentagon code, the standard example of a HaPPY code, as such operators are equivalent to correctable errors whose measurement cannot reveal any logical code information. Such correlation functions, while unphysical for finite $N$ CFTs, in fact accurately reflect the $N \to \infty$ limit of AdS/CFT (and more generally of *fixed area states*[14,15]), where all but the first term in (1) dominate. For the mutual information $I(A:B) = S_A + S_B - S_{A \cup B}$ of two subregions $A$ and $B$, this limit suggests that for distances between $A$ and $B$ much larger than their sizes, two-

point correlations exactly vanish by virtue of the bound

$$I(A:B) \geq \frac{\left( \langle \mathcal{O}_A \mathcal{O}_B \rangle - \langle \mathcal{O}_A \rangle \langle \mathcal{O}_B \rangle \right)^2}{2 ||\mathcal{O}_A||^2 ||\mathcal{O}_B||^2}, \qquad (2)$$

where $\mathcal{O}_{A,B}$ are arbitrary Hermitian operators acting on $A$ and $B$, respectively. In continuum AdS/CFT, physical correlation functions are restored by the subdominant bulk entropy term $S_a$ into (1). In the HaPPY code picture, a nonzero bulk term requires logical bulk states with long-distance entanglement, resembling the entanglement structure of bulk quantum fields. However, due the discretization of the bulk space, such entanglement is not resolved on sizes below the curvature radius (the size of a single tile), while in continuum AdS/CFT this sub-cutoff entanglement would become part of the area term[16]. As a result of this discretization, the code's resilience against small errors makes logical bulk states inaccessible to small boundary operators, causing their correlation functions to vanish.

Building holographic codes with physical boundary correlations thus seems to require breaking the encoding map $V : \mathcal{H}_{bulk} \to \mathcal{H}_{bdy}$ from an exact isometry (with $V^\dagger V = \mathbb{1}$) to an approximate one. Indeed, this has been argued to hold for codes describing continuum AdS/CFT as a consequence of the Reeh–Schlieder theorem for boundary quantum fields[17,18]. Tensor network models of holographic codes with approximate encoding have previously been constructed[19,20] and indeed allow for less constrained correlation functions that can decay polynomially. However, as they break exact bulk reconstruction, their features, e.g., state dependence of entanglement wedges, are difficult to analyze analytically. In addition, approximate encoding isometries cannot be directly implemented in terms of unitary gates in a quantum device. Rather than choosing an approximate encoding map, is it possible to construct holographic codes with physical correlation decay and approximate complementary recovery using an exact map? In Ref. 21 it was argued that any holographic code with local bulk reconstruction (i.e., tensor-by-tensor from boundary to bulk) can only achieve this by breaking the tiling symmetries and placing different tensors on different sites of the tiling.

In this work, we construct an exact tensor network code that preserves the tiling symmetries and still has the desired properties listed above. This requires only a mild relaxation of the local reconstruction property, compatible with expectations for a holographic model with weak gravitational back-reaction, that leads to a soft breaking of complementary recovery for any bipartition (shown in Fig. 1). The result is a tractable model of holography under quantum corrections, naturally producing physical correlation functions while relating holographic bulk states to the renormalization group (RG) flow of critical states on the boundary.

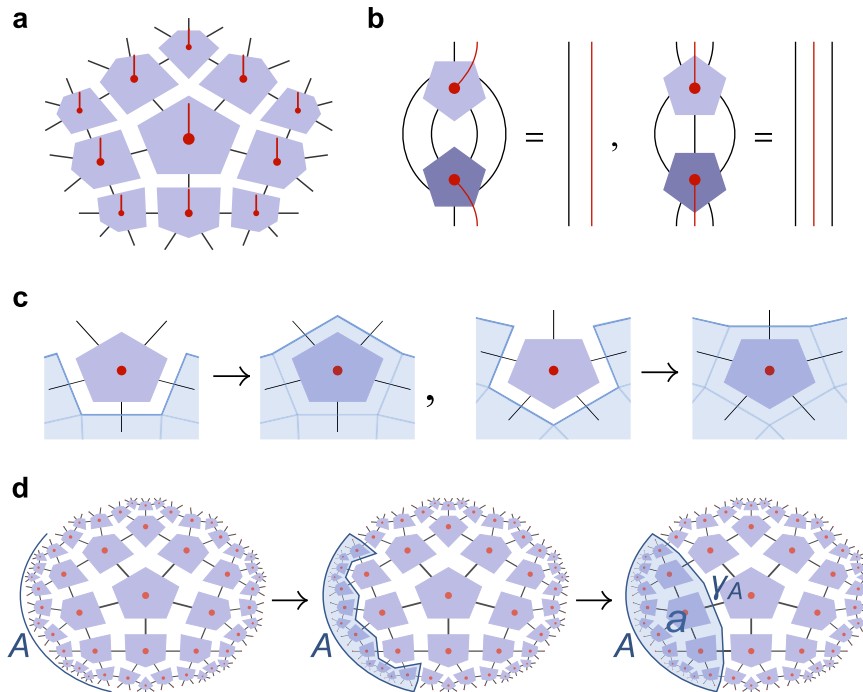

**Fig. 2 | Bulk reconstruction in HaPPY codes. a** A tensor network on a regular hyperbolic {4, 5} tiling, defining a map $V$ from logical bulk legs (red) to physical boundary legs (black). **b** Conditions of the form (4) for perfect pentagon tensors, the choice of which makes $V$ isometric and produces a HaPPY code. Dark-shaded tensors are conjugated. **c** The perfect tensor conditions define steps in a greedy algorithm for iteratively reconstructing the bulk from a boundary region $A$ (bulk legs not drawn). **d** The greedy algorithm terminates at a minimal cut $\gamma_A$, a discrete Ryu–Takayanagi surface. Within the wedge $a$ bounded by $A \cup \gamma_A$, logical operators can be isometrically mapped to $A$.

## Results

### Perfect holographic codes

We briefly review holographic codes based on perfect tensors on hyperbolic tilings. A regular $\{p, q\}$ tiling of $p$-gons, $q$ of which meet at each vertex, is hyperbolic if $pq > 2(p + q)$. In our notation, which follows Refs. 22,23 and is related to that of Ref. 12 by a $p \leftrightarrow q$ duality transformation, the vertices are associated with tensors that each have $q$ planar legs, with a loop of contractions between $p$ tensors around each tile. To form a bulk/boundary code, each tensor $T_{j,i_1,i_2,\ldots,i_q}$ has a bulk logical index $j$ representing the logical qudit of (bond) dimension $d$, and $q$ planar physical indices, each of dimension $\chi$, usually chosen as $\chi = d$. The tensor thus serves as an encoding isometry $V_T$ mapping a logical qudit state vector $|\psi\rangle$ to its physical encoding on $q$ sites,

$$V_T|\psi\rangle = \sum_{j=1}^{d} \sum_{i_1,\ldots,i_q=1}^{\chi} T_{j,i_1,\ldots,i_q}\langle j|\psi\rangle|i_1,\ldots,i_q\rangle \qquad (3)$$

For simplicity, we assume that the tensors follow the same symmetries as the (infinite) tiling, i.e., the same tensor is placed on each vertex and they are rotationally invariant under permutations of the physical indices, $T_{j,i_1,i_2,\ldots,i_q} = T_{j,i_q,i_1,\ldots,i_{q-1}}$. For the construction of holographic toy models proposed in Ref. 12, one further assumes that the tensor $T$ is *perfect*, defined as forming an isometry for any bipartition of its indices. More generally, a tensor $T$ is defined to be $k$-isometric (or equivalently, $k$-uniform[24]) if for any index bipartition into a set $S$ with $|S| = k$ indices and its complement $S^c$,

$$\sum_{S^c} T_{S,S^c} T_{S',S^c}^* \propto \delta_{S,S'}, \qquad (4)$$

where the sum runs over all indices in $S^c$, and $T_{S,S^c}$ is the tensor under the index bipartition. Under this definition, a tensor $T_{j,i_1,i_2,\ldots,i_q}$ is perfect

if it is $k$-isometric for any $k \leq \lfloor \frac{q+1}{2} \rfloor$. A quantum state represented by such a tensor thus appears maximally mixed to any observer with access to only $k$ sites or fewer. For perfect tensors, these states are given by absolutely maximally entangled (AME) states[24–29].

Given this definition, a HaPPY code is then defined as a tensor network of perfect tensors over a hyperbolic tiling, such that contraction over the physical indices between adjacent tiles/tensors results in an isometry from the bulk degrees of freedom to the physical degrees of freedom on the tiling boundary. As a regular tiling of the hyperbolic disk contains infinitely many tiles, this boundary can either be defined asymptotically or by some finite cutoff after a certain number of layers of tiles. The isometry condition for the bulk-to-boundary map $V$ follows immediately for most $\{p, q\}$ tilings, as the evaluation of $V^\dagger V$ can be decomposed into partial contractions between each perfect tensor $T$ and its complex conjugate $T^*$, resulting in expressions of the form (4). Graphically, this tile-by-tile reduction can be expressed by a greedy algorithm that iteratively pushes the tiling boundary into the bulk whenever this reduces the number of indices, corresponding to an application of the isometric map induced by any individual perfect tensor $T$ (see Fig. 2). An example where the greedy algorithm fails is the $\{7, 3\}$ tiling, where no local pushing (greedy step) reduces the number of indices. Indeed, a simple dimensional counting argument shows that a $\{7, 3\}$ tensor network of perfect tensors does not form an isometry and hence does not define a code. Fortunately, a non-trivial greedy algorithm can be applied to any perfect tensor network on a $\{p, q\}$ regular tiling with $q > 3$, such as the $\{4, 5\}$ hyperbolic pentagon code introduced in Ref. 12. HaPPY codes reproduce the AdS/CFT property of complementary recovery (Fig. 1) in that the greedy algorithm applied to a boundary region $A$ and its complement $A^c$ will terminate at the same cut through the tiling – the discretized Ryu-Takayanagi surface $\gamma_A$ – for almost all choices of $A$. The union of the bulk regions $a$ and $a^c$ that are recoverable from $A$ and $A^c$, respectively, then fills the entire bulk. As described above, exact

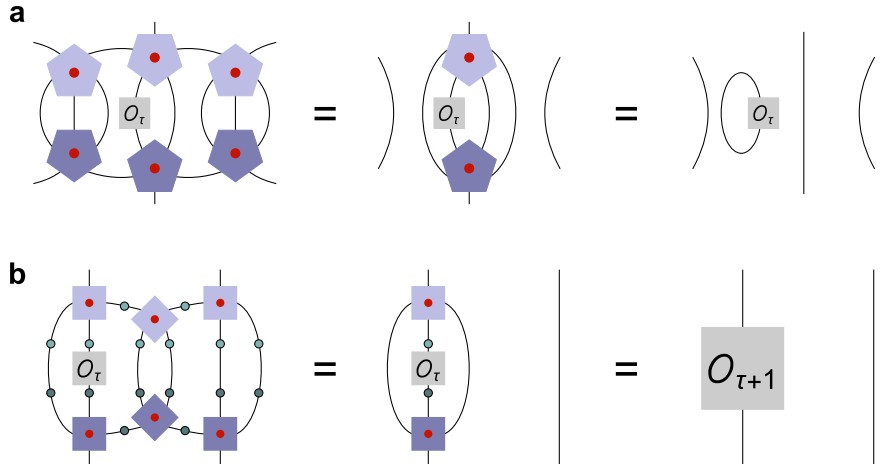

**Fig. 3 | Coarse-graining renormalization group step of a one-site operator. a** In the {4, 5} HaPPY code, any single-site operator $\mathcal{O}_\tau$ on layer $\tau$ of the tiling is a correctable error, and mapped to the identity on layer $\tau + 1$. **b** In the {5, 4} HTN code, $\mathcal{O}_\tau$ is generally mapped to a local $\mathcal{S}(\mathcal{O}_\tau) = \mathcal{O}_{\tau+1} \neq \mathbb{1}$, hence allowing for non-trivial single-site lattice primary operator with $\mathcal{O}_{\tau+1} \propto \mathcal{O}_\tau$. Tensors shaded in dark are conjugated, and red dots represent logical degrees of freedom, with legs suppressed.

complementary recovery is expected from AdS/CFT at $N \to \infty$, but creates problems for discrete holographic codes where code states are supposed to be related to CFTs and hence exhibit smoothly decaying correlation functions. A related problem arises when considering HaPPY codes from an RG perspective: In a tensor network representation of critical systems[30,31], the lattice version of a primary operator is an eigen-operator of the *scaling superoperator*, a map constructed from a single radial layer of tensor networks and its conjugate. However, the error-correcting properties of HaPPY codes are too strong to allow for non-trivial single-site operators to be preserved even under a single layer of the tensor network, as Fig. 3a shows. As a consequence, HaPPY codes cannot be related to any particular critical lattice theory with a spectrum of primary fields, a desirable feature of a discrete model of AdS/CFT.

## Hyperinvariant tensor networks

For the case of tensor networks without logical degrees of freedom, i.e., representations of holographic states rather than codes, the problem of non-trivial primary operators was resolved in Ref. 22 with the introduction of hyperinvariant tensor networks (HTN). This class of tensor networks has the same geometry as HaPPY codes, i.e., are constructed on a regular {*p*, *q*} hyperbolic tiling, with a *q*-leg vertex tensor *A* placed on each vertex. In addition, for each edge a 2-leg edge tensor *B* is contracted between two vertex tensors. For the choice of perfect *A* and a splittable $B = UU^\mathsf{T}$ with unitary *U* (in particular, $B = U = \mathbb{1}$), this construction just corresponds to a HaPPY code with the logical bulk projected onto a product state. However, HTNs allow for more general choices of *A* and *B* where layers of the tiling are both isometric and form super-operators with non-trivial spectra[22,23], resulting in a tensor network similar to the MERA[30] but with the geometry of HaPPY model. As we will now show, these conditions are general enough to also include holographic codes, i.e., HTNs whose vertex tensors have additional bulk legs and form an isometry from bulk to boundary. The main HTN model introduced in Ref. 22, based on a {7, 3} tiling with 3-leg vertex tensors, can be immediately excluded from such an extension: Just as in the case of HaPPY codes, adding a bulk leg to every vertex of a {7, 3} tiling leads to more degrees of freedom in the bulk than on the boundary, ruling out a bulk-to-boundary isometry. Here we assume for simplicity that the local Hilbert spaces of each bulk and boundary leg has the same dimension $\chi$. Upon closer inspection, one also finds that the HTN isometry conditions that would define a corresponding greedy algorithm cannot be fulfilled if the {7, 3} vertex tensors are associated with bulk qudits.

However, the second HTN model from Ref. 22, based on a {5, 4} tiling, is a viable candidate for an HTN code: Here the two isometry conditions (known as *multitensor constraints* in[22]), the first for a single vertex tensor and the second for two neighboring ones, can be extended into reconstruction steps for bulk qudits, which we show in Fig. 4. The 4-leg tensor *A* is promoted to a 5-leg tensor $A'$, while the *B* tensor remains a fixed unitary matrix that does not depend on the bulk state. Note that the isometry conditions differ from the perfect tensor conditions of HaPPY codes (see Fig. 2): Firstly, they do not require the $A'$ tensor to be perfect but only 1-isometric for all bipartitions and 2-isometric for those where the smaller side of the bipartition contains the logical leg. Secondly, bulk reconstruction is possible for {*p*, *q*} codes with even *q*, unlike HaPPY codes where a single logical site must be reconstructable from $\lfloor \frac{q}{2} \rfloor$ physical indices. Thirdly, they include the isometric constraints where combinations of neighboring tensors act as larger isometries recovering more than one logical site, in contrast to the HaPPY code's greedy algorithm acting only on one tensor at a time.

## HaPPY and hyperinvariant codes

Before giving an explicit solution to the HTN code conditions, we first explore the consequences of such conditions for bulk reconstruction. As noted above, tensors $A'$ that fulfill the isometric constraints of Fig. 4d for a specific *B* tensor can also be perfect. In that case, given a {*p*, *q*} tiling with odd *q*, bulk reconstruction would proceed as in the HaPPY code, where the union $a \cup a^c$ of the greedy wedges of *A* and $A^c$ fills the entire bulk. However, we specifically wish to construct HTN codes with non-perfect tensors, whose super-operators have non-trivial spectra. In that case, we find that the bulk region that can be reconstructed for all states in the code space is strictly smaller than the HaPPY wedge. As we further find that the size of the reconstructable region is generally dependent on the bulk state, we will refer to it as the *reconstruction wedge* in analogy to work on state-dependent bulk reconstruction in continuum AdS/CFT[32,33]. That the HTN reconstruction wedge can be significantly smaller than the HaPPY wedge is a direct consequence of utilizing non-perfect tensors. Specifically, the HTN isometry conditions in Fig. 4d and their generalizations to other {*p*, *q*} tilings induce a greedy algorithm that is unable to produce connected bulk wedges from disconnected boundary regions. This is because these conditions, unlike in the HaPPY code, act as isometries only on neighboring legs. This property is similar to that of block-perfect tensors (which represent planar maximally-entangled (PME) states[34]) that have been previously considered in the context of holography[35]. A particular feature of HTN codes that follows from

Hyperinvariant tensor network (HTN)

**a**

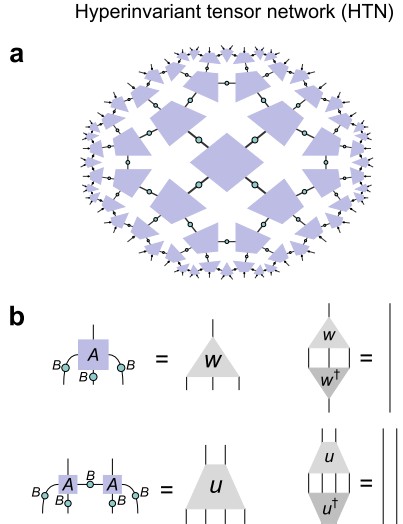

HTN bulk–boundary code

**c**

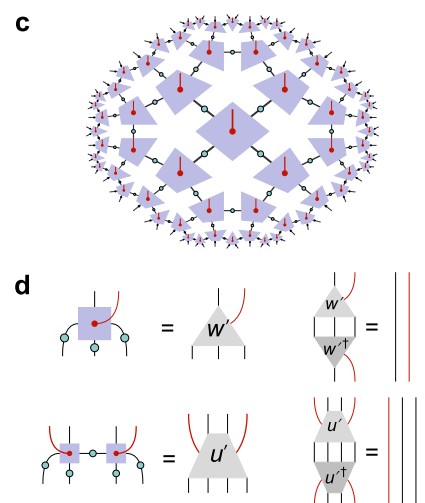

**Fig. 4 | Construction of a hyperinvariant tensor network (HTN) code. a** The original HTN construction from Ref. 22,23 for a {5, 4} hyperbolic tiling; vertices and edges are associated with $A$ and $B$ tensors, respectively. **b** The isometry constraints for one- and two-vertex combinations of $A$ and $B$. With these constraints, each radial layer of tensors acts isometrically, in the manner of an RG transformation. **c** Implanting an additional leg on each $A$ tensor yields a bulk Hilbert space, with the tensor network acting as a bulk-boundary map. **d** The updated isometry constraints (with logical legs in red) turn the RG step into a bulk reconstruction step. Despite the $A$ tensors not being necessarily perfect as in the HaPPY code (Fig. 2), the tensor network acts as an exact bulk-to-boundary isometry.

non-perfectness is that (guaranteed) complementary recovery is broken even between a connected boundary region $A$ and its complement $A^c$. An example of this is shown in Fig. 1b: Even though $A$ and $A^c$ fill the entire boundary, their respective reconstruction wedges $a$ and $a^c$ exclude a strip of bulk sites that stretches all the way between the endpoints $\partial A$. While in HaPPY codes such residual bulk regions can take up $O(1)$ bulk sites, for HTN codes their number scales with the radial cutoff and is divergent for the infinite tiling. That this property is a consequence of non-perfect tensors can be seen in the example of a single tensor, as shown in Fig. 5a, b: While perfect tensors allow logical reconstruction on either $A$ or $A^c$ (whichever is larger), for the HTN case with a merely 1-isometric vertex tensor, some bipartitions rule out exact reconstruction both on $A$ and $A^c$. For the entire tiling, this property then ensures a strip-shaped residual region around the minimal cut $\gamma_A$ in the tiling, as tensors that are half connected to $a$ and $a^c$ cannot be reached from either side by the greedy algorithm. This behavior was previously observed in the original HTN model without bulk legs[36].

The breakdown of exact complementary recovery changes the form of the entanglement entropy $S_A$ of a boundary region $A$. In the HaPPY code, complementary recovery is realized via an isometry $V_a : \mathcal{H}_a \otimes \mathcal{H}_{\gamma_A} \to \mathcal{H}_A$ from the logical qudits in the entanglement wedge $a$ and the physical qudits on the legs cut by the RT surface $\gamma_A$ to the boundary region $A$. This isometry can be constructed by simply contracting the tensors in $a$. Evaluating the von Neumann entropy $S[\rho_A]$ of the reduced density matrix $\rho_A = \mathrm{tr}_{A^c} \rho$ then yields the sum of two terms: The entropy of the reduced logical bulk state $\tilde{\rho}_a = \mathrm{tr}_{a^c} \tilde{\rho}$ and the entanglement along $\gamma_A$, which is composed of maximally entangled pairs due to the perfect tensor condition. This leads to the expression

$$S_A = |\gamma_A| \log \chi + S_a, \qquad (5)$$

where $|\gamma_A|$ is the number of edges of the minimal cut $\gamma_A$, $\chi$ the bond dimension, and $S_a = S[\tilde{\rho}_a]$ the bulk entropy. This expression clearly resembles the continuum AdS/CFT formula (1) without $O(G)$ corrections, i.e., in the semi-classical limit.

For HTN codes, exact complementary recovery is broken and (5) no longer holds. Instead, $S_A$ now depends non-trivially on three contributions: The bulk entropy of the reconstruction wedge $a_r$, the non-

maximal entanglement along the bulk cut $\gamma'_A$ (with $\partial a_r = A \cup \gamma'_A$), and the entanglement mediated between $\gamma'_A$ and $\gamma'_{A^c}$ through the bulk residual region $r$, all of which are state-dependent. However, for a bulk state with negligible entanglement between $r$ and $r^c = a_r \cup a_r^c$ we expect an approximate form

$$S_A \simeq S_{\gamma'_A}[V_r^\dagger \tilde{\rho}_r V_r] + S_{a_r}. \qquad (6)$$

Here the first term is the entanglement mediated from $\gamma'_A$ to $\gamma'_{A^c}$ through the residual region $r$, which is state-dependent on $\tilde{\rho}_r = \mathrm{tr}_a \mathrm{tr}_{a^c} \tilde{\rho}$, with $V_r$ being the isometry from logical qudits in $r$ to $\gamma'_A \cup \gamma'_{A^c}$. As this term scales with the length of the minimal surface $\gamma_A$ and is bounded by $|\gamma_A| \log \chi$, we identify it as the state-dependent area term in AdS/CFT under quantum corrections[7,10], with an example of such state dependence given below.

Without complementary recovery, non-trivial RG transformations become possible. As we show in Fig. 3b, a local operator $\mathcal{O}_\tau$ on level $\tau$ of the HTN tiling is coarse-grained into a local operator $\mathcal{S}(\mathcal{O}_\tau) = \mathcal{O}_{\tau+1}$ on level $\tau + 1$, where $\mathcal{S}$ is the scaling superoperator formed from one radial layer of the tiling and its conjugate. Unlike HaPPY codes, $\mathcal{O}_{\tau+1}$ is generally not the identity, and one can find lattice primary operators $\mathcal{O}_\alpha$ for which

$$\mathcal{S}(\mathcal{O}_\alpha) = s^{\Delta_\alpha} \mathcal{O}_\alpha, \qquad (7)$$

where $s$ is the scaling factor of the tiling and $\Delta_\alpha$ the scaling dimension of $\mathcal{O}_\alpha$[22]. The property of finding non-trivial lattice primary operators is directly related to the code properties: Applying the scaling superoperator is equivalent to pushing an operator $\mathcal{O}_\tau$ through a radial layer of tensors, i.e., replacing it with another operator $\mathcal{O}_{\tau+1}$ that acts on different physical sites while acting identically on the codespace of the logical qudits in that layer. For any state vector $|\bar{\psi}\rangle$ in the codespace, we thus require

$$\mathcal{O}_\tau |\bar{\psi}\rangle = \mathcal{O}_{\tau+1} |\bar{\psi}\rangle \leftrightarrow \langle \bar{\psi} | \mathcal{O}_{\tau+1}^{-1} \mathcal{O}_\tau | \bar{\psi}\rangle = 1. \qquad (8)$$

For a HaPPY code built from perfect tensors, the only operators $\mathcal{O}_\tau$ and $\mathcal{O}_{\tau+1}$ (each with support on a different single site) that fulfill this condition are the identity, as all two-site errors are correctable. This

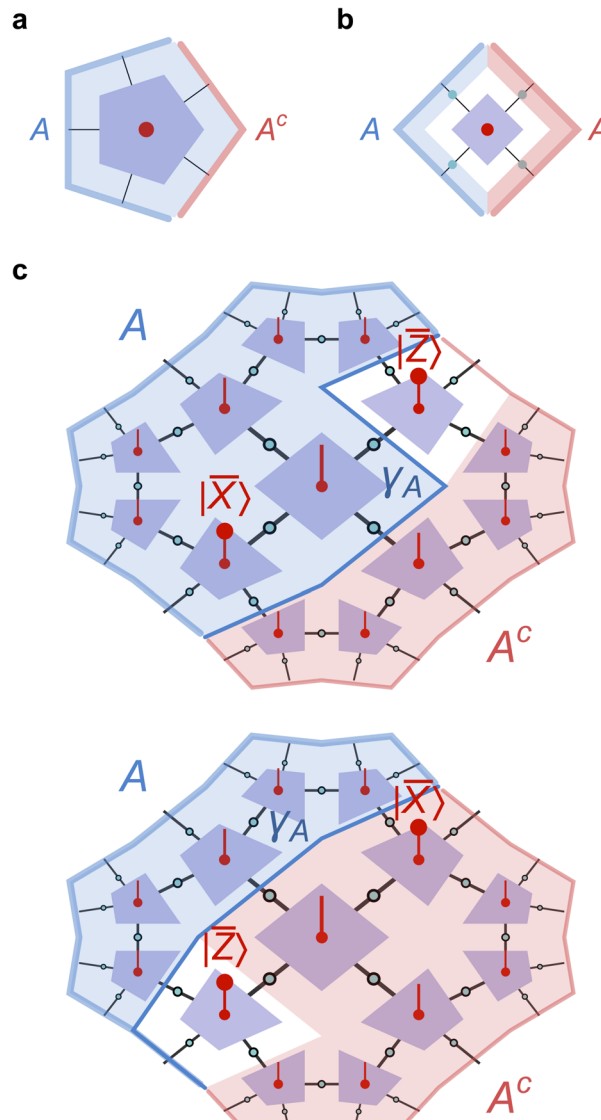

**Fig. 5 | Bulk reconstruction in HTN codes. a** For perfect tensors, any index bipartition into regions $A$ and $A^c$ leads to a reconstruction of the logical index (red dot) on either $A$ or $A^c$. **b** For the non-perfect tensors used in the HTN code, bipartitions exist for which neither $A$ nor $A^c$ are sufficient for reconstruction. **c** For a patch of the $\{5, 4\}$ ququart HTN code with $A'$ and $B$ tensors given by (9) and (12), we can show state-dependent reconstruction explicitly: For the given boundary regions $A$ and $A^c$, the central bulk ququart cannot be state-independently reconstructed on either. But given local projections of the neighboring bulk ququart on eigenstates of the logical ququart Paulis $\bar{X}$ and $\bar{Z}$, it can be fully reconstructed on either $A$ or $A^c$. While the RT surface $\gamma_A$ bounding the reconstructible region changes, its length $|\gamma_A|$ remains constant.

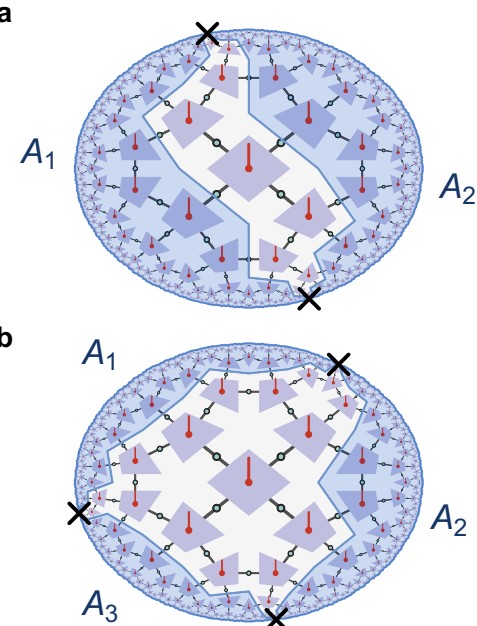

**Fig. 6 | Effect of site erasures on bulk reconstruction. a** In an HTN code, erasure of two boundary sites (black crosses) bisects the remaining boundary into two disjoint parts $A_1$ and $A_2$, with the reconstruction wedge of $A_1 \cup A_2$ (shaded blue) being disjoint as well, separated by a strip-like region. **b** For three single-site erasures, the reconstruction wedge of $A_1 \cup A_2 \cup A_3$ leaves out a larger region in the center of the bulk. Thus even small boundary operations can have an effect on information deep in the bulk for part of the codespace.

forces all two-point functions between non-trivial single-site operators to vanish. For HTN codes, where the vertex tensors are 1-isometric along planar legs, only single-site errors are correctable, and hence two-point functions are generally nonzero and non-trivial primaries exist.

The behavior of $n$-point correlation functions in HTN codes is closely related to the correctability of $n$-point errors for $n > 2$, as well. Consider the single-site erasures for the $\{5, 4\}$ HTN code shown in Fig. 6: Erasure of two single boundary sites prevents guaranteed reconstruction of a strip of logical sites on a geodesic between the two erasures, similar to the bipartition case of Fig. 1b. This implies that two-point correlation functions are heavily distance-dependent

and that long-range correlators can probe logical information deep in the bulk. For the three erasures in Fig. 6b, the residual bulk region becomes even wider, consisting of the bulk volume enclosed by the discrete geodesics $\gamma_{A_k}$ of the three disconnected boundary regions $A_k$. This drastically changes the code properties of HTN codes when compared to HaPPY codes: In HaPPY codes, bulk qudits remain resilient even against high-weight boundary errors as long as those are sufficiently sparse, a property know as *uberholography*[37]. On the contrary, HTN codes offer no such protection of logical information deep in the bulk against sparse errors with small support on the boundary. This comes with the caveat of state dependence: Boundary reconstruction beyond the reconstruction wedge may be possible for a semi-classical subspace of the logical Hilbert space whose protection against boundary errors matches that of the HaPPY code.

Earlier work on state dependence in tensor network codes considered modifications of the HaPPY code by inclusion of a black hole, i.e., a random tensor of high bond dimension (close to a perfect tensor at high bond dimension) whose logical leg, representing the black hole microstates, exhibits a variable dimension: For certain boundary regions this setting allows for approximate reconstruction of bulk operators when the reconstruction involves pushing them through the black hole tensor, which is possible if the mixed state of the black hole has small entropy (i.e., small logical dimension in the random tensor model). It has been argued that such state-dependent, approximate recovery is an essential feature of continuum AdS/CFT[32]. Beyond mixed bulk states involving black holes, one further expects to find state dependence for pure bulk states even close to the vacuum state, i.e., with only little backreaction to the AdS geometry[33]. State dependence and approximate recovery are thus unavoidable features of holographic codes, even if one restricts the bulk code space to low-energy perturbations around the vacuum. For a boundary bipartition into regions $A$ and $A^c$, this state dependence should appear as a splitting of the logical code space into *α-blocks*

each corresponding to a different Ryu–Takayanagi surface $\gamma_A$ separating the entanglement wedges $a$ and $a^c$, representing different bulk geometries in superposition[14,15]. It is this aspect in which HTN codes generalize HaPPY codes, as they exhibit some state dependence for pure states. An example is shown in Fig. 5c for the $\{5, 4\}$ tiling, using the explicit code that we introduce in the next section: Here, The central logical qudit can be reconstructed from a boundary region $A$ comprising half of the boundary only for a subspace of the full logical bulk in which two logical qudits next to the center are projected onto eigenstates of the logical Pauli operators $\bar{X}$ or $\bar{Z}$ (here denoted as $|\bar{X}\rangle$ and $|\bar{Z}\rangle$, with an index for the specific eigenstate suppressed). For a different projection, the central logical qudit can be reconstructed on $A^c$ instead. We can thus interpret the logical subspaces given by these projections as different $\alpha$-blocks. While this allows for state-dependent bulk reconstruction with different $\gamma_A$ in each subspace, we see in our example that the length $|\gamma_A|$, i.e., the number of physical indices that $\gamma_A$ cuts across, remains constant. This is no coincidence: As shown recently, all stabilizer codes have trivial area operators[38]. This means that the area term of the entanglement entropy, when written in an algebraic decomposition along $\alpha$-blocks, is a block-independent scalar depending only on the choice of $A$. Hence, HTN codes exhibit the maximum amount of state dependence that stabilizer codes allow.

Given such state dependence, we can identify tensor network analogs of fixed-area states with flat entanglement spectra for any boundary bipartition into connected regions $A$ and $A^c$, thus acting like states in a HaPPY code. In our explicit HTN code example, this state corresponds to a projection onto local $|\bar{X}\rangle$ eigenstates (for $d$-dimensional qudits, these are $d$ states spanning the logical Hilbert space). These $d^N$ states $|\bar{X}\rangle^{\otimes N}$ thus form the fixed-area basis of the bulk Hilbert space. Each fixed-area state has trivial superoperator spectra as in Fig. 3b, behaving like classical AdS vacua without any bulk modes. Conversely, superpositions of these states can be interpreted as excited bulk modes with back-reaction, leading to non-trivial boundary correlation functions. Therefore, to describe the boundary ground state of a critical theory we start with a bulk product state $|\bar{\psi}_{\text{gnd}}\rangle^{\otimes N}$ that is not a fixed-area state, and extract the operator spectrum of the boundary theory from the resulting superoperators. Low-energy excitations within this theory are then given by bulk states that locally deviate from $|\bar{\psi}_{\text{gnd}}\rangle$.

## An explicit code construction

We now give an example of an HTN code that fulfills the HTN constraints for the $\{5, 4\}$ geometry as visualized in Fig. 4d. This example uses qudits with local dimension $d = 4$ (ququarts), where each encoding tensor $A'$ represents an error-detection code spanned by the logical states[26]

$$|\bar{0}\rangle = \frac{1}{2}(|0000\rangle + |1111\rangle + |2222\rangle + |3333\rangle), \tag{9a}$$

$$|\bar{1}\rangle = \frac{1}{2}(|0123\rangle + |1230\rangle + |2301\rangle + |3012\rangle), \tag{9b}$$

$$|\bar{2}\rangle = \frac{1}{2}(|0202\rangle + |1313\rangle + |2020\rangle + |3131\rangle), \tag{9c}$$

$$|\bar{3}\rangle = \frac{1}{2}(|0321\rangle + |1032\rangle + |2103\rangle + |3210\rangle), \tag{9d}$$

which is stabilized by the ququart Pauli generators $XXXX$, $IZZ^2Z$, and $ZZ^2ZI$. The logical operators of this code can be represented as $\bar{X} = IXX^2X^3$ and $\bar{Z} = IIZ^3Z$ or any cyclic permutations of the tensor products, as the code is completely invariant under a rotation of the physical qubits. Equivalently, the tensor $A'$ is invariant under cyclic

permutation of the planar indices,

$$A'_{j,i_1,i_2,i_3,i_4} = A'_{j,i_4,i_1,i_2,i_3}, \tag{10}$$

where $j$ is the logical index and $i_k$ are the planar ones. The projection onto any logical state results in a 4-ququart state that is 1-isometric on the planar legs, which implies that no logical information can be extracted from any single physical ququart. Thus (9) defines a $[\![4, 1, 2]\!]_4$ code that can detect a single-site error. We now check the isometry condition for $w'$ and $u'$ in Fig. 4(d). The $w'$ condition only depends on the $A'$ tensor as long as $B$ is unitary, and given (10), it suffices to evaluate the condition from the logical index $j$ and the first planar index $i_1$ to the remaining planar ones. Expressing $w'^\dagger w'$ in index notation, we find

$$\sum_{i_2,i_3,i_4 = 0}^{3} T_{j,i_1,i_2,i_3,i_4} T^\star_{j',i'_1,i_2,i_3,i_4} \propto \delta_{j,j'}\delta_{i_1,i'_1}. \tag{11}$$

In order to also fulfill the isometry (or rather unitary) condition for $u'$, we choose the $B$ tensor to be the ququart Hadamard matrix,

$$B = \frac{1}{2}H_4 = \frac{1}{2}\begin{pmatrix} 1 & 1 & 1 & 1 \\ 1 & -1 & 1 & -1 \\ 1 & 1 & 1 & -1 \\ 1 & -1 & -1 & 1 \end{pmatrix}, \tag{12}$$

which is both symmetric and unitary. We find that this choice of $A'$ and $B$ leads to $u'$ being unitary, but omit the full expression for the sake of clarity. Another valid solution for $B$ that uses the same $A'$ is given by the 4-dimensional quantum Fourier transform. As the tensor $A'$ is generally non-perfect along its planar indices, it follows that the respective superoperators exhibit non-trivial spectra. However, any projection onto an eigenstate of $\bar{X}$, such as

$$|\bar{X}_1\rangle = \frac{1}{2}(|\bar{0}\rangle + |\bar{1}\rangle + |\bar{2}\rangle + |\bar{3}\rangle), \tag{13}$$

will produce a tensor that is block-perfect on its remaining four planar legs. This is one possible choice for the state $|\bar{X}\rangle$ discussed above in the context of state-dependent bulk reconstruction, where $|\bar{Z}\rangle$ can be chosen as any of the four basis states $|\bar{Z}_k\rangle \equiv |\bar{k}\rangle$. The proof of the specific reconstruction shown in Fig. 5c for this code is described in the Methods section 4 using operator pushing techniques.

The 4-qudit HTN code also produces non-trivial superoperator spectra for certain bulk (product) states. The 1-site superoperator $\mathcal{S}$ shown in Fig. 3b, for example, has four Hermitian eigenoperators for the bulk state $\alpha|\bar{0}\rangle + \sqrt{1 - \alpha^2}|\bar{2}\rangle$ with $0 \leq \alpha \leq 1$, two of which have positive eigenvalue: The identity $\mathbb{1}$ with eigenvalue 1, and the operator

$$\mathcal{O} = \frac{1}{\sqrt{1+\lambda^2}}\begin{pmatrix} 1 & 0 & \lambda & 0 \\ 0 & 1 & 0 & \lambda \\ \lambda & 0 & -1 & 0 \\ 0 & \lambda & 0 & -1 \end{pmatrix} \tag{14}$$

with eigenvalue $\lambda = \sqrt{2\alpha\sqrt{1 - \alpha^2}}$. As $0 \leq \lambda \leq 1$, the norm of this operator generally decays under coarse-graining $\mathcal{O} \to \mathcal{S}(\mathcal{O})$, as expected from an RG transformation of a primary operator. Using the scale factor $s = 2 + \sqrt{3}$ for the $\{5, 4\}$ tiling, we can thus associate the operator $\mathcal{O}$ with a scaling dimension $\Delta = -\log_s\lambda$[22].

A similar analysis can be performed for any choice of tensors that form an HTN code. A systematic search for critical lattice models that can be described by HTN codes, as well as the precise relationship between code properties and critical spectra, will be an interesting subject for future work. We already note here that such models will

break boundary translation invariance as a consequence of the tiling symmetries. Most instances will therefore not have a well-defined CFT continuum limit but will fall into the more general class of *quasiperiodic CFTs*[39].

## Discussion

A number of previous models have been proposed as approximate holographic codes in the past, the earliest being random tensor networks at finite bond dimension[19,40]. The resulting codes are approximate in their encoding isometry, making it difficult to study their properties analytically. Subsequent tensor network codes with exact encoding isometries but approximate complementary recovery have also been proposed, using tensors that alternate between perfect and non-perfect ones[20,21]. Such models have less symmetry than HaPPY codes and break complementary recovery only for some bipartitions, whereas it is softly broken in HTN codes for any bipartition. Similarly, two-point correlation functions in HTN codes are generically nonzero, whereas they must vanish e.g., in the hybrid holographic code of Ref. 20 for small operators acting on one of the perfect tensors. HTN codes thus have more symmetrical features and retain the structure of stabilizer codes while only introducing small violations of locality during bulk reconstruction that are consistent with the expectation of small quantum gravity effects. Following this logic, a holographic code representing AdS/CFT with strong quantum gravity contributions can be built by requiring more intricate isometric constraints, representing the breakdown of a semi-classical geometry on which the bulk information is located.

As HTN codes break exact complementary recovery, it will be interesting to explore whether restrictions on HaPPY codes regarding fault-tolerant logical operations using boundary transversal gates[41] still apply in this new setting. A holographic code relating local boundary to local bulk time evolution, potentially realizable by

an HTN code, would also have a number of useful features regarding non-local quantum computation[42,43]. Another interesting question is whether HTN codes also support non-stabilizer subsystem codes such as those constructed in Ref. 20, and how this affects state dependence.

While HTN and HaPPY codes share the same tensor network geometry, we saw that their resilience against erasure errors is somewhat different, with HTN codes suppressing the effect of boundary operations on logical qudits deep in the bulk approximately through an RG process, rather than guaranteeing uberholographic recoverability. In ongoing work, we show that more general classes of HTN codes built from 2-isometric states beyond the {5, 4} tiling can be constructed[44]. There we also analyze their quantum error-correction properties and discuss applications for practical quantum computing.

## Methods

The hyperinvariant code construction is purely analytical and can be checked by explicit calculation on paper or using computer algebra systems. We now verify our central claim of state-dependent recovery of logical bulk qudits (shown in Fig. 5c) using a graphical proof based on operator pushing for the ququart HTN code defined by the $[\![4, 1, 2]\!]_4$ code (9) and the $B$ tensor (12). For this purpose, we show that the logical algebra generated by the logical ququart Pauli operators $\bar{X}$ and $\bar{Z}$ can be reconstructed state-dependently on either the region $A$ or its complement $A^c$, given projections of the neighboring two ququarts on eigenstates of $\bar{X}$ or $\bar{Z}$. Specifically, one starts with a representation of $\bar{X}$ or $\bar{Z}$ in terms of physical Pauli operators on the internal (contracted) indices of the tensor, and then applies operator pushing: By applying stabilizers, we can remove Pauli operators on one physical index at the cost of adding new ones elsewhere, all while leaving the tensor invariant. For $A'$ tensors, whose logical leg is projected onto an eigenstate $|\bar{X}\rangle$ or $|\bar{Z}\rangle$ of a ququart Pauli operator, applying the respective logical

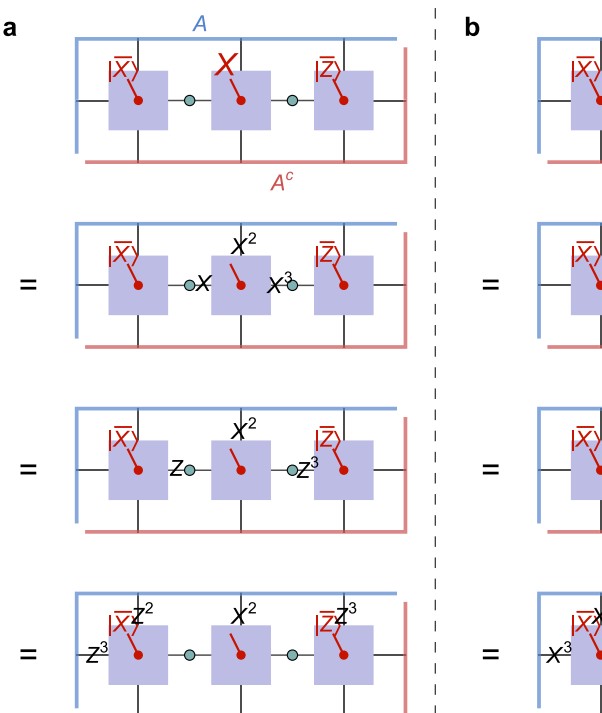
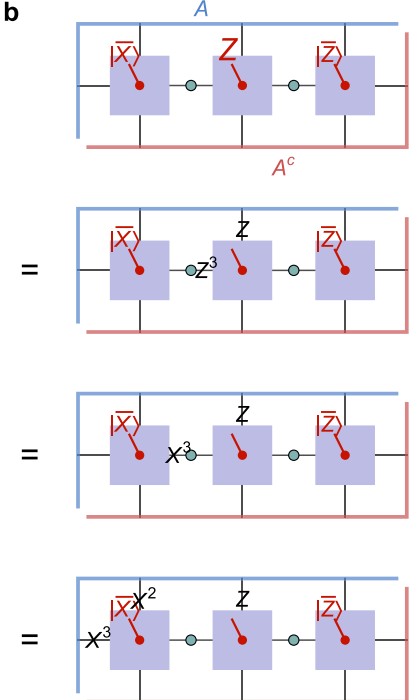

**Fig. 7 | Graphical proof of the state-dependent reconstruction of the central ququart in Fig. 5c. a** The ququart Pauli operator $X$ is represented by the logical operator $\bar{X} = XX^2X^3I$. After pushing the Pauli operators acting on internal indices through the $B$ tensors (green dots), which exchanges $X \leftrightarrow Z$, we apply the stabilizer $(ZZ^2ZI)^3 = Z^3Z^2Z^3I$ on the left side and $\bar{Z}^3 = (Z^3ZII)^3 = ZZ^3II$ on the right, resulting in a physical operator represented only on boundary region $A$. **b** Similarly, we can push $\bar{Z} = Z^3ZII$ to the left, apply $\bar{X}^3 = (XX^2X^3I)^3 = X^3X^2XI$, and again arrive at a physical operator on $A$.

operator (or any power thereof) is also an invariant operation. We can also move Pauli operators past $B$ tensors by using the identities $XH_4 = H_4Z^T$ and $ZH = HX^T$, which exchange ququart $X$ and $Z$ (note the transpose, as the direction in which the operator acts is reversed).

The part of the tensor network in Fig. 5c that is relevant for our proof consists of a block of three $A'$ tensors and the two $B$ tensors between them that would form the bulk residual region given no restriction on the bulk Hilbert space. The actual operator pushing steps of this proof are shown in Fig. 7 for the setup in which the left $A'$ tensor is projected onto $|\bar{X}\rangle$, and the right one onto $|\bar{Z}\rangle$. We find that both $\bar{X}$ and $\bar{Z}$ acting on the central ququart can be represented as Pauli operators acting purely on the subregion $A$. By symmetry, swapping the two projections results in the opposite scenario, where representation only on $A^c$ is possible, corresponding to the two settings in Fig. 5c. We also confirmed these results numerically, finding in the first scenario that the mutual information between the logical index and $A$ is $\log 4$, while it vanishes with regards to $A^c$, and vice versa in the second scenario. This proves genuinely state-dependent reconstruction: For different logical subspaces, the central logical ququart lies in different entanglement wedges.

## Data availability
The authors declare that the data supporting the findings of this study are available within the paper.

## Code availability
The Mathematica code used to verify the analytical calculations is available from the corresponding author on request.

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

## Acknowledgements
We would like to thank Chris Akers, Charles Cao, Philippe Faist, Jens Eisert, David Elkouss, and John Preskill for helpful discussions and comments. M.S. and S.F. are grateful for financial support from the Intel Corporation. A.J. is supported by the Simons Collaboration on It from Qubit, the US Department of Energy (DE-SC0018407), and the Einstein Research Unit "Perspectives of a quantum digital transformation".

## Author contributions
The paper was prepared jointly by the authors. M.S. developed the tensor network code which is the main result of this paper. S.F. and A.J. provided project guidance, with S.F. contributing to quantum code aspects and AJ developing the connection to holography.

## Funding

## Competing interests
The authors declare no competing interests.
