## [Peer Review File · Nature Communications]

Holographic Codes from Hyperinvariant Tensor NetworksREVIEWER COMMENTS

Reviewer #1 (Remarks to the Author):

The authors explore a QECC adaptation of the hyper-invariant tensor network (HTN) as a holographic toy model. They find a symmetric construction of a code that is exact unitarily constructible and is able to reproduce power-law correlation functions. None of the existing models is able to satisfy all above criteria. The code inherits the capacity to reproduce power-law correlation from HTN for certain implementations of the tensors fixed to certain bulk states. Bulk reconstruction is state-dependent, which is very similar to [32] and [19]. Unlike the HaPPY pentagon code, the boundary seems to be able to produce non-sparse power law using only the bulk state variations, which is an interesting feature of their construction.

However, I am also puzzled by the authors' claim that the code reproduces back-reactions. The most convincing evidence that the authors provide for back-reaction is entropy change. However, this alone is insufficient for the following reasons:

1. Complementary recovery is heavily broken in their model, whereas it is expected to hold in AdS/CFT for a pure state, even when the state is heavy like a large (pure state) black hole.
2. Although back-reaction can deform the minimal surface, the authors implicitly assume that this deformation is justified by the boundary entropy change because the global bulk state is pure. However, in models with broken subsystem code complementary recovery, such entropy change need not be come from the area operator contributions. See especially (5.7)-(5.9) of [11].
3. More recently, [<https://arxiv.org/abs/2306.14996>] claims that stabilizer codes are insufficient in capturing back-reaction. Since the authors' example with ququart code and Fourier gate is a stabilizer code, their findings seem to be at odds with constraints on holographic stabilizer codes.

Overall I find the HTN code to be a novel and valuable class of models that fills the gaps left by existing holographic toy models. It also seems to provide a convenient class of ansatzes that is more naturally tied to variational methods for CFTs. However, I do not find convincing the authors' claims on quantum gravity corrections and back-reaction as they

seem to be at odds with known properties of AdS/CFT. I recommend publication if the authors can either address the apparent conflicts or remove their claims on back-reaction.

Reviewer #2 (Remarks to the Author):

It is of great interest to understand how to make holographic error-correcting codes more "realistic" in various ways. The manuscript under review contributes to this task by showing that the previously introduced hyper-invariant codes display a breakdown of complementary recovery in precisely the way expected when relaxing away from the semiclassical ($N \rightarrow \infty$) limit. This represents an advance in the field and deserves to be published. I don't have many comments on the technical core of the paper, which is an extension of the HTN and scaling superoperator discussions in [20,21], but I think some aspects of the introduction which relate the AdS/CFT discussion to error correction could be improved--see the detailed comments below.

Introduction:

"deep connections to fundamental theory": it's not quite clear what 'fundamental' means here. Arguably, a QEC is more fundamental than a holographic CFT in the sense that it requires much less data to describe -- just a Hilbert space and a code subalgebra, as opposed to the entire set of OPE coefficients required to define the CFT. Maybe "high-energy" or "gravitational" would be better than "fundamental."

"conveniently made finite by a choice of discretization and radial cutoff": one should also add that the gauge has been (partially) fixed by the choice of timeslice even before this discretization.

"This is because the high-energy sector of the CFT is dual to strong bulk perturbations of the AdS background (i.e., black hole states) that are outside of the semi-classical approximation." This is correct but (I think) misleading. It's true that the low-energy sector of the CFT corresponds to perturbations about the AdS vacuum. But the QEC picture of AdS/CFT doesn't just work for the empty AdS geometry, it is supposed to work for *every

semiclassical geometry that is asymptotically AdS with a fixed value of the CC*. That is, there are multiple distinct good semi-classical approximations, but the sentence in the Introduction I've quoted only applies to one particular of these approximations, the one around the AdS vacuum. So I think talking about the high- and low-energy sectors of the CFT is not the cleanest way to discuss this, because (for example) a CFT state that corresponds to a large black hole backgrounds could be very high energy and yet perfectly geometric. Instead it's probably cleaner to talk, e.g., in terms of identity block dominance.

"The logical space of semi-classical AdS states" As per the above discussion, clearer to say "the logical space of states close to a fixed semiclassical background" or something similar.

"For a bipartition $H = H_A \otimes H_C$ of the boundary Hilbert space" - it should be noted that a priori a CFT does not cleanly factorize in this way, so one needs to e.g. impose a UV cutoff, gauge-fix, etc. This is a *boundary* discretization and thus distinct from the bulk field discretization referred to earlier in the paragraph (though they should of course have some relation)

Eq. (1): As stated earlier $G \sim 1/N^2$, so it's a bit odd to say that this equation includes $O(N^0)$ corrections when it also includes corrections subleading to that.

Reviewer #1 (Remarks to the Author):

The authors explore a QECC adaptation of the hyper-invariant tensor network (HTN) as a holographic toy model. They find a symmetric construction of a code that is exact unitarily constructible and is able to reproduce power-law correlation functions. None of the existing models is able to satisfy all above criteria. The code inherits the capacity to reproduce power-law correlation from HTN for certain implementations of the tensors fixed to certain bulk states. Bulk reconstruction is state-dependent, which is very similar to [32] and [19]. Unlike the HaPPY pentagon code, the boundary seems to be able to produce non-sparse power law using only the bulk state variations, which is an interesting feature of their construction.

We agree with the Reviewer's summary of our results.

However, I am also puzzled by the authors' claim that the code reproduces back-reactions. The most convincing evidence that the authors provide for back-reaction is entropy change. However, this alone is insufficient for the following reasons:

- 1. Complementary recovery is heavily broken in their model, whereas it is expected to hold in AdS/CFT for a pure state, even when the state is heavy like a large (pure state) black hole.*
- 2. Although back-reaction can deform the minimal surface, the authors implicitly assume that this deformation is justified by the boundary entropy change because the global bulk state is pure. However, in models with broken subsystem code complementary recovery, such entropy change need not be come from the area operator contributions. See especially (5.7)-(5.9) of [11].*
- 3. More recently, [<https://arxiv.org/abs/2306.14996>] claims that stabilizer codes are insufficient in capturing back-reaction. Since the authors' example with ququart code and Fourier gate is a stabilizer code, their findings seem to be at odds with constraints on holographic stabilizer codes.*

We thank the Reviewer for pointing out this important caveat: It is correct that stabilizer codes cannot have non-trivial area operators, as proven in the work by C. Cao [arXiv:2306.14996] just after submission of our manuscript. It is also true that a change in boundary entanglement for different pure bulk states can be attributed to the "bulk term" of the (algebraic) entanglement entropy, rather than a change in the RT surface. To show that our HTN codes truly exhibit state dependence, we have made several changes to our manuscript:

1. Fig. 5(c) has been modified to show a genuine example of state dependence with HTN codes, where the reconstruction of one logical qubit can be performed on either a boundary region A or its complement, depending on the state of its neighbors.
2. Fig. 6 (previously showing state-dependent entanglement entropy) has been removed, as the state-dependent reconstruction example of Fig. 5 provides a more convincing argument for quantum gravity corrections, as noted by the Reviewer.

3. The latter part of Sec. IIC (p. 7) has been completely rewritten, describing the results of Fig. 5(c) in detail and stressing the connection to “fixed-area states” with flat entanglement spectra, which we called “HaPPY-like states” in the original manuscript.
4. The notation of Sec. IID has been slightly updated to conform with Fig. 5(c), and references to Fig. 6 have been removed.
5. We have added an Appendix with a new Fig. 7 to the manuscript, which provides a graphical proof of the state-dependent reconstruction shown in Fig. 5(c), using the explicit HTN code from Sec. IID.

Crucially, our holographic (stabilizer) code circumvents Cao’s no-go result by allowing for state-dependent entanglement wedges whose length remains state-independent, thus producing trivial area operators.

Overall I find the HTN code to be a novel and valuable class of models that fills the gaps left by existing holographic toy models. It also seems to provide a convenient class of ansatzes that is more naturally tied to variational methods for CFTs. However, I do not find convincing the authors' claims on quantum gravity corrections and back-reaction as they seem to be at odds with known properties of AdS/CFT. I recommend publication if the authors can either address the apparent conflicts or remove their claims on back-reaction.

We thank the Reviewer for their positive feedback, and believe that our manuscript changes regarding state dependence have convincingly demonstrated a rudimentary form of back-reaction in HTN codes.

Reviewer #2 (Remarks to the Author):

It is of great interest to understand how to make holographic error-correcting codes more "realistic" in various ways. The manuscript under review contributes to this task by showing that the previously introduced hyper-invariant codes display a breakdown of complementary recovery in precisely the way expected when relaxing away from the semiclassical ($N \rightarrow \infty$) limit. This represents an advance in the field and deserves to be published. I don't have many comments on the technical core of the paper, which is an extension of the HTN and scaling superoperator discussions in [20,21], but I think some aspects of the introduction which relate the AdS/CFT discussion to error correction could be improved--see the detailed comments below.

We thank the Reviewer for their positive appraisal of our work and for providing helpful comments.

Introduction:

"deep connections to fundamental theory": it's not quite clear what 'fundamental' means here. Arguably, a QEC is more fundamental than a holographic CFT in the sense that it requires much less data to describe -- just a Hilbert space and a code subalgebra, as

opposed to the entire set of OPE coefficients required to define the CFT. Maybe "high-energy" or "gravitational" would be better than "fundamental."

We agree that the term “fundamental” is ambiguous here, and have replaced the sentence with: “The field of quantum error correction [...] has deep connections to high-energy theory and quantum gravity.”

"conveniently made finite by a choice of discretization and radial cutoff": one should also add that the gauge has been (partially) fixed by the choice of timeslice even before this discretization.

To clarify that choosing a time-slice imposes a (gauge) restriction, we have changed this sentence to: “Counter-intuitively, the number of degrees of freedom of a bulk field on a semi-classical AdS background, when restricted onto a chosen time-slice at time t and made finite via discretization and a radial cutoff, is *smaller* than that of the boundary CFT state.”

"This is because the high-energy sector of the CFT is dual to strong bulk perturbations of the AdS background (i.e., black hole states) that are outside of the semi-classical approximation." This is correct but (I think) misleading. It's true that the low-energy sector of the CFT corresponds to perturbations about the AdS vacuum. But the QEC picture of AdS/CFT doesn't just work for the empty AdS geometry, it is supposed to work for every semiclassical geometry that is asymptotically AdS with a fixed value of the CC. That is, there are multiple distinct good semi-classical approximations, but the sentence in the Introduction I've quoted only applies to one particular of these approximations, the one around the AdS vacuum. So I think talking about the high- and low-energy sectors of the CFT is not the cleanest way to discuss this, because (for example) a CFT state that corresponds to a large black hole backgrounds could be very high energy and yet perfectly geometric. Instead it's probably cleaner to talk, e.g., in terms of identity block dominance.*

The Reviewer has raised an important point: The code picture of continuum AdS/CFT is supposed to hold for any semi-classical bulk state, not just low-energy ones. However, tensor network models of holography are arguably even more restrictive, as the tensor network geometry is fixed even deep in the bulk and unable to describe perturbations with strong back-reaction (and high energy). For example, modeling a black hole with the HaPPY code requires the modification of tensors in the bulk, changing the underlying code. Another reason to restrict a holographic code to low energies is to preserve an approximately isometric encoding, which black hole states can violate. To address this point, we've updated the sentence to read: “This is because considering only bulk states on a fixed semi-classical geometry imposes a restriction on the boundary Hilbert space, leaving out CFT states that are dual to a non-geometrical bulk or contain strong back-reaction (such as black hole states).”

"The logical space of semi-classical AdS states" As per the above discussion, clearer to say "the logical space of states close to a fixed semiclassical background" or something similar.

The sentence now reads: "As a result, the AdS/CFT dictionary becomes an (approximately) isometric code between bulk and boundary: The logical space of states associated with a semi-classical AdS geometry are encoded in a *code subspace* of the boundary CFT states."

*"For a bipartition $H = H_A \otimes H_{A^c}$ of the boundary Hilbert space" - it should be noted that a priori a CFT does not cleanly factorize in this way, so one needs to e.g. impose a UV cutoff, gauge-fix, etc. This is a *boundary* discretization and thus distinct from the bulk field discretization referred to earlier in the paragraph (though they should of course have some relation)*

We've added a comment clarifying this point: "For a bipartition $H = H_A \otimes H_{A^c}$ of the boundary Hilbert space (which is finite-dimensional due to the bulk discretization and cutoff), [...]"

Eq. (1): As stated earlier $G \sim 1/N^2$, so it's a bit odd to say that this equation includes $O(N^0)$ corrections when it also includes corrections subleading to that.

To avoid confusion, we've rephrased this statement to: "Explicitly including $O(N^0)$ corrections [...]"

REVIEWERS' COMMENTS

Reviewer #1 (Remarks to the Author):

My thanks to the authors for their response. They have addressed the comments I made in the review and I recommend for publication.